# Retinal cholesterol metabolism is perturbated in response to experimental glaucoma in the rat

Elise Léger-Charnay[1], Ségolène Gambert[1,2], Lucy Martine[1], Elisabeth Dubus[1], Marie-Annick Maire[1], Bénédicte Buteau[1], Tristan Morala[1], Vincent Gigot[1], Alain M. Bron[1,3], Lionel Bretillon[1], Elodie A. Y. Masson[1]*

1 Centre des Sciences du Goût et de l'Alimentation, AgroSup Dijon, CNRS, INRAE, Université Bourgogne Franche-Comté, Dijon, France, 2 Laboratoire de Biochimie Médicale, Plateforme de Biologie Hospitalo-Universitaire, Dijon, France, 3 Département d'Ophtalmologie, Centre Hospitalo-Universitaire de Dijon, Dijon, France

* elodie.masson@inrae.fr

**Data Availability Statement:** The relevant data for this study is publicly available at the Data INRAE database at https://doi.org/10.15454/SR56ZY.

## Abstract

Alterations of cholesterol metabolism have been described for many neurodegenerative pathologies, such as Alzheimer's disease in the brain and age-related macular degeneration in the retina. Recent evidence suggests that glaucoma, which is characterized by the progressive death of retinal ganglion cells, could also be associated with disruption of cholesterol homeostasis. In the present study we characterized cholesterol metabolism in a rat model of laser-induced intraocular hypertension, the main risk factor for glaucoma. Sterol levels were measured using gas-chromatography and cholesterol-related gene expression using quantitative RT-PCR at various time-points. As early as 18 hours after the laser procedure, genes implicated in cholesterol biosynthesis and uptake were upregulated (+49% and +100% for HMG-CoA reductase and LDLR genes respectively, vs. naive eyes) while genes involved in efflux were downregulated (-26% and -37% for ApoE and CYP27A1 genes, respectively). Cholesterol and precursor levels were consecutively elevated 3 days post-laser (+14%, +40% and +194% for cholesterol, desmosterol and lathosterol, respectively). Interestingly, counter-regulatory mechanisms were transcriptionally activated following these initial dysregulations, which were associated with the restoration of retinal cholesterol homeostasis, favorable to ganglion cell viability, one month after the laser-induced ocular hypertension. In conclusion, we report here for the first time that ocular hypertension is associated with transient major dynamic changes in retinal cholesterol metabolism.

## Introduction

Among its multiple roles in the body, cholesterol is a component of cell membranes, crucial for the maintenance of their structure and fluidity. Neurons are especially dependent on cholesterol since they need to build up large amounts of membrane for their axons, dendrites and synapse growth. Well-balanced inputs and outputs of cholesterol, tuned to fit with demand,

**Funding:** This work was supported by grants from the Institut National de la Recherche Agronomique; the Conseil Régional Bourgogne, Franche-Comté (PARI grant); the FEDER (European Funding for Regional Economical Development); the Fondation de France/Fondation de l'œil; the Ministère de l'Enseignement Supérieur, de la Recherche et de l'Innovation; the Université de Bourgogne Franche-Comté; and the Nouvelle Société Française d'Athérosclérose. The funders had no role in study design, data collection and interpretation, or the decision to submit the work for publication. The funders had no role in study design, data collection and analysis, decision to publish, or preparation of the manuscript.

**Competing interests:** The authors have declared that no competing interests exist.

**Abbreviations:** 24S-OHC, 24(S)-hydroxycholesterol; 27-COOH, 3β-hydroxy-5-cholestenoic acid; 27-OHC, 27-hydroxycholesterol; ApoE, Apolipoprotein E; Brn3a, Brain-specific homeobox/POU domain protein 3a; Brn3b, Brain-specific homeobox/POU domain protein 3b; CD68, Cluster of differentiation 68; CYP27A1, Cytochrome P450 family 27 subfamily A member 1; CYP46A1, Cytochrome P450 family 46 subfamily A member 1; GC-FID, Gas chromatography coupled to flame ionization detector; GFAP, Glial fibrillary acidic protein; Gusb, β-glucuronidase; HMGCR, HMG-CoA reductase; Iba1, Ionized calcium binding adaptor molecule 1; IOP, Intraocular pressure; LDLR, Low density lipoprotein receptor; LRP1, Low density lipoprotein receptor-related protein 1; LXR, Liver X receptor; RBPMS, RNA-binding protein with multiple splicing; RGC, Retinal ganglion cell; RPE, Retinal pigment epithelium; RQ, Relative quantification; SR-BI, Scavenger receptor class B member 1; SREBP2, Sterol regulatory element-binding protein 2; Thy-1, Thymocyte differentiation antigen 1; TNFα, Tumor necrosis factor alpha; TRADD, TNFR1-associated death domain protein.

are therefore fundamental for the healthy functioning of nervous tissues, such as the brain and retina. The retina is made of several layers of neurons in close proximity to macroglial and microglial cells, as well as blood vessels. In this tissue, cholesterol originates from local *de novo* biosynthesis as well as uptake of cholesterol-rich blood lipoproteins from the systemic circulation [1]. Müller cells, the major macroglia in the retina, may be cholesterol providers for adult neurons since they have been shown to express HMG-CoA reductase (HMGCR), the rate limiting enzyme of cholesterol synthesis, as well as apolipoprotein E (ApoE) and ABCA1 transporters, involved in cholesterol export [1–4]. The retinal pigment epithelium (RPE), lying between the neuro-retina and the choriocapillaris, is involved in blood lipoprotein uptake via the expression of low density lipoprotein receptor (LDLR) and scavenger-receptors (SR-BI, BII and CD36) at the level of its basal membrane [5]. Müller and RPE cells could then export cholesterol in HDL-like particles [6] that can be subsequently captured by surrounding neurons expressing LDL family receptors (LDLR/LRP1). To be eliminated from the retina, cholesterol is excreted into the circulation via reverse cholesterol transport or via conversion into oxysterols. 24(S)-hydroxycholesterol (24S-OHC) and 5-cholestenoic acid (27-COOH), synthesized respectively by the retinal ganglion cell (RGC)-specific CYP46A1 and the more ubiquitous CYP27A1, are major routes for retinal cholesterol output [7]. Cholesterol metabolism in the retina, as a summary of the data gathered from the literature, is illustrated in Fig 1.

Cholesterol in nervous tissues is gaining more attention since studies are accumulating showing that perturbation of its metabolism is associated with various neurodegenerative diseases. In the brain, it is indeed well documented that cholesterol disturbances are linked to the Smith-Lemli Opitz syndrome or the Nieman-Pick type C disease, and likely also to Alzheimer's and Huntington diseases [8, 9]. In the retina, during age-related macular degeneration, cholesterol has been shown to accumulate in the aging Bruch's membrane and to be a major component of the lipid-rich deposits typical of the disease [7]. Additionally, the discovery of variants in cholesterol-related genes (*Abca1*, *Apoe*, . . .) as risk factors for age-related macular degeneration strengthens the importance of cholesterol in disease development [10, 11]. Finally, a single nucleotide polymorphism in the gene coding for CYP46A1 was found to be associated with a higher risk for developing glaucoma, a neurodegenerative disease of the retina and the optic nerve [12].

Glaucoma is the leading cause of irreversible blindness worldwide [13]. This pathology encompasses a group of several conditions characterized by the progressive death of the RGCs. Intraocular pressure (IOP) is the main and only modifiable risk factor for glaucoma [14]. There is still no curative treatment and a better understanding of the biochemical mechanisms involved in the physiopathology of the disease is crucial in order to develop new neuroprotective therapies. Interestingly, elevated IOP has been shown to induce an increase in the levels of CYP46A1 and its product 24S-OHC, in an *ex-vivo* model of glaucoma [15]. Transient increases in CYP46A1 expression and 24S-OHC levels, in association with glial activation and inflammation, have also been reported in the rat retina following laser-induced elevation of IOP [16]. However, there are currently no published data providing a comprehensive characterization of cholesterol metabolism *in vivo* in response to glaucoma-related stress. In the present study, we report the modification of various aspects of cholesterol metabolism, associated with RGC death, inflammation and glial activation in the rat retina after laser-induced elevation of IOP. We used a time course approach to capture changes at early and later stages.

## Materials and methods

### Animals

Experiments were conducted on eight-week male Sprague-Dawley rats (Janvier, Le Genest Saint Isle, France). Animals were housed under controlled temperature (22±1˚C), hygrometry

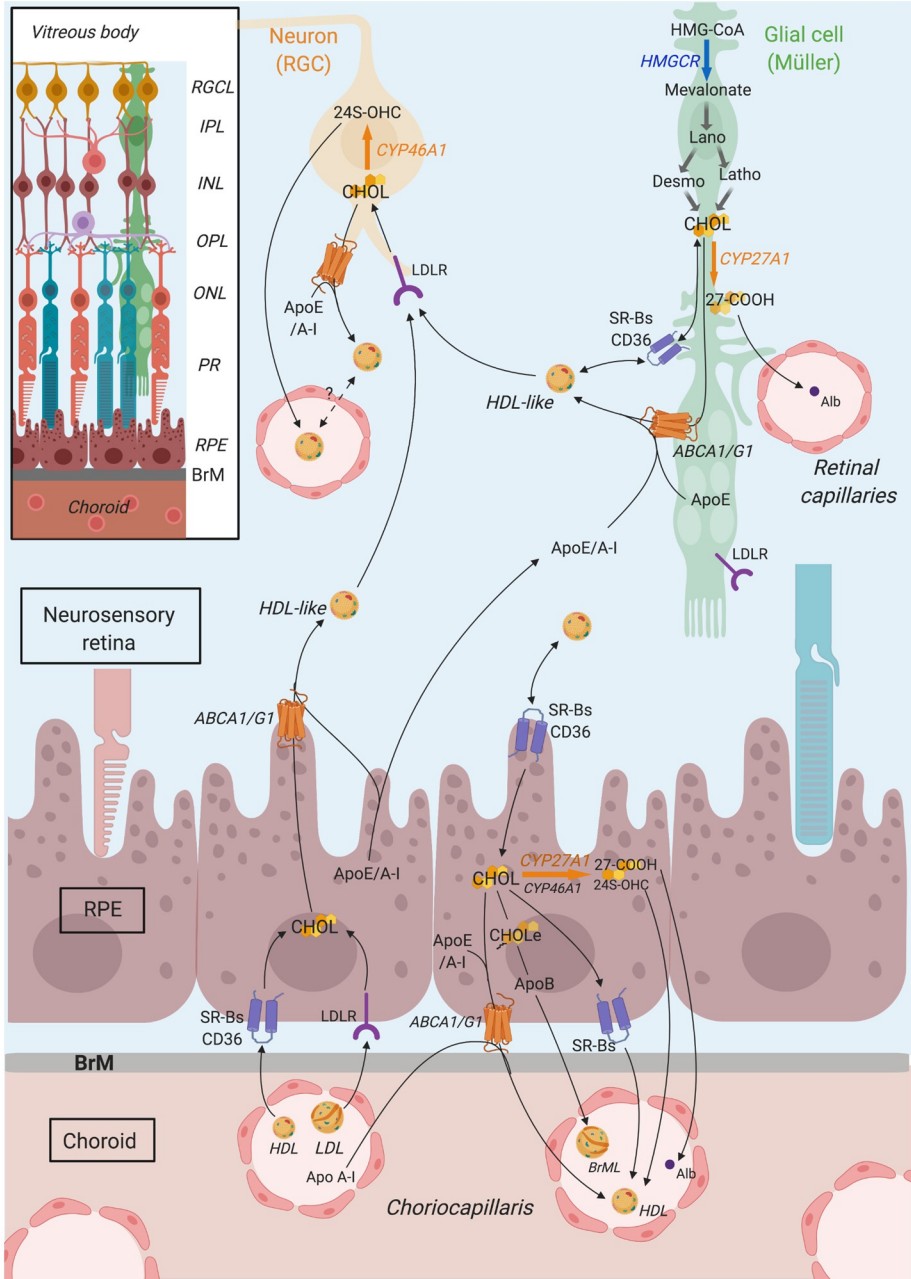

**Fig 1. Cholesterol metabolism in the retina.** Cholesterol in the retina originates from a local biosynthesis, that likely occurs primarily in glial cells which express HMG-CoA reductase (HMGCR) enzyme, and from the uptake of blood lipoproteins by retinal pigment epithelium (RPE) cells, expressing LDLR family receptors (LDLR) and scavenger receptors (SRBs, CD36). Within the retina, cholesterol is thought to be transported in HDL-like particles, rich in Apo E/AI, secreted by RPE and glial cells, thanks to ABC transporters. HDL-like could be subsequently captured by neighboring cells, especially neurons, or could potentially cross retinal capillaries to reach the systemic circulation and be eliminated. Moreover, RPE cells secrete cholesterol-rich particles, including Bruch's membrane particles (BrML) at their basal membrane, that reach the circulation via choroidal capillaries. Cholesterol output also involves its conversion into oxysterols, the main two being 24(S)-hydroxycholesterol (24S-OHC) and 3ß-hydroxy-5-cholestenoic acid (27-COOH), produced by CYP46A1 and CYP27A1 enzymes, respectively. CYP46A1 is almost exclusively expressed in retinal ganglion cells (RGCs). CYP27A1 is more ubiquitously expressed in the retina, including likely photoreceptors and Müller cells. *CHOL: cholesterol, Desmo: Desmosterol, Latho: Lathosterol, CHOLe: Cholesterol esters, Alb: Albumin. RGCL: RGC layer, IPL: Inner plexiform layer, INL: Inner nuclear layer, OPL: Outer plexiform layer, ONL: Outer nuclear layer, PR: Photoreceptors. BrM: Bruch's membrane.* Created in Biorender.com. and reprinted under a CC BY 4.0 license, with permission from Biorender, original copyright 2021.

(55–60%) and light conditions (50lux, 12h/12h dark/light cycles). Food and water were provided *ab libitum*. All procedures were approved by the local ethic committee (Comité d'Ethique de l'Expérimentation Animale Dijon Grand-Campus, University of Burgundy, Dijon, France) and the French ministry of Higher Education and Research. They were also in accordance with the Association for Research in Vision and Ophthalmology (ARVO) statement for the use of animals in Ophthalmic and Vision Research.

## Intraocular pressure elevation by laser photocoagulation

At day 0, animals were anesthetized with an intraperitoneal injection of a ketamine (50mg/kg, Imalgène ®1000, Merial, Lyon, France) and xylazine (6mg/kg, Rompun ® 2%, Bayer, Puteaux, France) mix. The right eye of rats received local anaesthesia (Tetracaïne 1%, Théa, Clermont-Ferrand, France) and was subjected to laser photocoagulation, according to a protocol adapted from Salinas-Navarro and Levkovitch-Verbin [17, 18]. Briefly, episcleral veins (20–30 spots), limbus (50–60 spots) and trabecular meshwork (50–60 spots) were photocoagulated with a 532nm laser (Vitra-Quantel Medical, Clermont-Ferrand, France) connected to a slit lamp. The laser was set at 400mW for 0.5s with a beam diameter at 260 μm. Left eyes, which were not subjected to the laser procedure, were considered as contralateral eyes. Rats which were not subjected to the laser procedure on either eye were considered as naive. Thirty minutes after the end of the procedure, animals were injected with atipamezole (0.1 mg/kg, Narcostop®, Ceva, Libourne, France) to counteract the sedative effect of xylazin. Eye drops (Optive fusion™, Allergan, Annecy, France) were topically applied during and after surgery to prevent excessive dryness of the cornea. IOP was monitored, with a rebound tonometer (Tonolab®, Tiolat, Helsinki, Finland), 18 hours after the laser procedure and then at days 1, 3, 14 and 30. To facilitate animal handling and avoid excessive stress during the measurement, rats were slightly sedated with isoflurane (Vetflurane®, Virbac, Carros, France).

## Gene expression analysis by Taqman™ real time polymerase chain reaction

Rats were sacrificed by inhalation of increased concentrations of $CO_2$. Eyes were dissected on ice, retinas were snap frozen in liquid nitrogen without delay and then stored at -80˚C. RNA isolation was performed using the Nucleospin® RNA/protein kit (Macherey-Nagel, Hoerdt, France) according to manufacturer's instructions. Reverse transcription (RT) was performed with the Quantitec® Reverse Transcription Kit (Qiagen, Courtaboeuf, France) with 500ng RNA per reaction. Quantitative PCR was carried out with 10ng cDNA, TaqMan® Gene Expression Master Mix and TaqMan® probes (TaqMan® Gene Expression-Single tube assays, Thermofisher scientific, Life technologies, Courtaboeuf, France) (for primer references see the S1 Table). Plates were run on a StepOnePlus™ Real-Time PCR system (Applied biosystem, Thermofisher scientific) with the following cycle profile: denaturing at 95˚C for 20 s followed by 40 cycles at 95˚C for 1 s and 60˚C for 20 s. β-glucuronidase (*Gusb*) expression was used as a reference to normalize cDNA amounts. Relative quantification (RQ) values were calculated by the StepOne v2.3 software from automatic threshold and baseline. Data were analysed using the Livak's method ($2^{-\Delta\Delta CT}$) [19].

## Protein expression analysis by Western Blot

Retinal proteins were isolated along with RNA using the Nucleospin® RNA/protein kit (Macherey-Nagel, Hoerdt, France) according to manufacturer's instructions. Protein samples were resuspended in PSB-TCEP buffer and boiled 5 min. HMGCR and GFAP proteins were separated using a precast stain free 4–15% gel (BioRad stain free technology®, BioRad, Hercules, CA, USA). Then, proteins were transferred to a nitrocellulose membrane using the

TransBlot system (BioRad). Membranes were blocked 1 hour at room temperature in 5% fat-free dry milk in 1X PBS 0.1% tween 20. Incubation with primary antibodies (mouse monoclonal anti-GFAP [GA-5], Abcam, Cambridge, UK, 1/1000; rabbit monoclonal anti-HMGCR [EPR1685(N)], Abcam, 1/1000) was performed overnight at 4°C. After 3 washes in 1X PBS 0.1% tween 20, membranes were incubated 1 hour at room temperature with the appropriate peroxidase-conjugated secondary antibody (dilution 1/1000). After 3 washes in 1X PBS 0.1% tween 20, detection of the chemiluminescent signal was performed using the Western Lightning® Plus-ECL, PerkinElmer, Waltham, MA, USA) and a Bio-Rad Molecular Imager Chemidoc XRS+ system. Normalization was performed using total proteins quantified with the stain free technology using the BioRad Image Lab software.

## Immunohistofluorescence on flat-mounted retinas and frozen sections

Rats were deeply anesthetized with an intraperitoneal injection of pentobarbital (1 mL/kg, Dolethal, Vetoquinol®, Lure, France). Animals were perfused transcardially with a saline solution composed of 0.9% NaCl to wash out the blood prior to perfusion with 400mL of 4% paraformaldehyde (PFA) in 0.1 M phosphate buffer used as a fixative, for 20 minutes.

**Flat-mounted retinas.** Eyes were collected and post-fixed by immersion in 4% PFA for one hour at 4°C. Retinas were dissected, cut radially to allow them to flatten, and post-fixed again for one hour at 4°C. After gentle brushing to remove vitreous body, retinas were incubated overnight at 4°C with Brn3a antibody (1:100, mouse monoclonal sc-8429, Santa Cruz Biotechnologies, Heidelberg, Germany) diluted in blocking buffer (PBS, 2% normal goat serum (Dako, Courtaboeuf, France), 2% triton X100). Retinas were washed three times in 0.5% triton-PBS and incubated with anti-mouse Alexa 488 antibody (1:500, A11-001, Thermofisher scientific, Illkirch, France) in blocking buffer, for 2 hours at room temperature. After two washes in 0.5% triton- PBS, retinas were rinsed in PBS, and then mounted vitreal side up in anti-fading solution (FluoMounting Medium, Dako).

**Retinal cross-sections.** Eyes were post-fixed by immersion in 4% PFA for 30 minutes at 4°C. Cornea and lens were removed and eyecups were post-fixed one additional hour at 4°C. After progressive cryoprotection in 15–30% sucrose in 0.1M phosphate buffer, eyecups were embedded in optimal cutting temperature compound (Tissue-Tek®, Sakura, Finetek, Torrance, CA) and frozen in liquid nitrogen. Samples were stored at -80°C before and after cryosectioning. Immunolabelling was performed as follows. Sections (10μm) were rehydrated for 5 min in PBS, permeated and saturated in 1% BSA-PBS, 0.1% Triton X100, 0.05% tween ® 20, for 1 hour at room temperature, then incubated overnight at 4°C in primary antibody (mouse anti-Glial fibrillary acidic protein (GFAP), AYZ280, Interchim, 1:100; rabbit anti-Ionized calcium binding adaptor molecule 1 (Iba1), PA5-27436, Thermofisher scientific, 1:200; MCA341R, 1:200; rabbit anti-cleaved-Caspase 3a, 9661S, Cell Signaling Technology, 1:500; mouse anti-Brn3a, MAB 1585, Merck Millipore, 1:125) diluted in saturating buffer. Sections were rinsed three-times in PBS and incubated with the corresponding species-specific Alexa fluor-conjugated secondary antibody (goat anti-mouse 488, A11001; goat anti-mouse 594, A11005; goat anti-rabbit 488, Thermofisher scientific; 1:500) in saturating buffer, for 1 hour at room temperature. Cell nuclei were visualized with DAPI (4', 6-diamidino-2-phenylindole), incubated with secondary antibodies. After washing, sections were mounted in anti-fading solution.

**Image acquisition and analysis.** Images were acquired with a SP8 Leica confocal laser-scanning microscope connected to Leica Application Suite X (LasX) software. For flat-mounted retinas, images were captured as multi-frame acquisitions, using a hybrid detector, x20 objective and 1.12 zoom with a 2048x2048 pixel resolution. Focus correction was manually

applied on strategic points to guarantee a clear view of the RGC layer in each tile. Images were combined automatically into single tiled image by LasX software. The total number of RGC was counted on the whole retina with a script developed in our laboratory, on ImageJ 1.52g. Automatic counting was supplemented by manual counting when necessary. For retinal sections, images were captured using a hybrid detector, x40 objective with 1024x1024 resolution.

## Sterol quantification by gas chromatography

Retinas were collected as described for the gene expression analysis and stored at -80˚C until used. Total lipids were extracted according to the Moilanen and Nikkari protocol [20]. Briefly, proteins were precipitated with a solution of chloroform-methanol (1:1, v/v). A chloroform-acidic NaCl solution was mixed with the supernatant and total lipids were collected in the lower chloroform phase. For sterol and oxysterol quantification, assays were carried out following the procedure described previously [4]. In brief, total lipids from one retina were submitted to alkaline hydrolysis and material that could not be saponified, containing sterols, was extracted in chloroform. Cholesterol was quantified from 1/10 of the total non-saponified material, after trimethylsilyl ether derivation and addition of 5α-cholestane as a standard, by gas chromatography coupled to a flame ionization detector (GC-FID) (HP4890A, Hewlett-Packard, DB-5MS column, Agilent, Santa Clara, CA). Cholesterol precursors and oxysterols were quantified using the remaining 9/10 of the non-saponified fraction, after purification on silica columns (Supelco®, Sigma, St Quentin-Fallavier, France), derivation to trimethylsilyl ethers and addition of deuterated standards ([26,26,26,27,27,27-$^2$H$_6$] desmosterol, [25,26,26,26,27,27,27-$^2$H$_7$] lathosterol, [25,26,26,26,27,27,27-$^2$H$_7$] 24S-OHC), by gas chromatography coupled to mass spectrometry (GC-MS) (5973N, Agilent; DB-5MS column, Agilent, Santa Clara, CA). Sterols were identified in SCAN mode according to their specific spectra and retention times defined by unlabelled standards. Sterols were quantified in SIM mode with specific ions. For 27-COOH quantitation, [25,25,26,26,26-$^2$H$_5$] 27-COOH was added to total lipids extracted from one retina. Samples were not submitted to alkaline hydrolysis but directly purified on silica columns, according to the protocol mentioned above for cholesterol precursor and oxysterol quantitation. After diazomethane and trimethylsilyl ether derivations, 27-COOH was quantified using GC-MS.

## Statistical analysis

Statistical analyses were run with GraphPad Prism6 software (GraphPad Software, San Diego, USA). All the tests were two-tailed. A paired t- test was applied, at each experimental time point, to compare IOP in laser-treated *versus* contralateral eyes. An unpaired t-test was used to compare IOP in laser-treated or contralateral eyes *versus* naive eyes (n = 25 animals minimum per group). Because standard deviation (SD) was different between laser-treated and naive, the Welch's correction was applied to these groups. Regarding gene expression analyses, sterol quantification and RGC counting, we performed Wilcoxon matched-pair tests to compare laser-treated *versus* contralateral eyes, and Mann-Whitney tests to compare laser-treated or contralateral eyes *versus* naive ones (n = 6–8 in each group for the gene expression analysis, immunolabelling, sterol quantification and RGC counting). The significance threshold was set at 5%.

# Results

## Characterization of the experimental model of glaucoma

The IOP was monitored regularly over the experimental time period following the laser procedure (S1 Fig). IOP of contralateral and naive eyes remained steady at 9 ± 0.2 mmHg contrary

to the IOP of the laser-treated eyes which was clearly increased as early as 18h after the procedure and was maintained until day 3 (34 ± 1.4 mmHg, p<0.0001 *vs* contralateral or naive). Thereafter the IOP of laser-treated eyes progressively returned to baseline, remaining significantly higher than control eyes during 21 days.

This ocular hypertension was associated with an activation of macroglial cells as shown by a strong GFAP staining on the retinal sections of laser-treated eyes (Fig 2A). In naive eyes, GFAP staining was low and restricted to the nerve fibre layer. Following laser treatment, activated Müller cells appeared clearly at 18 hours, and especially prominently from 3 days, crossing the retina from the nerve fibre layer to the outer nuclear layer. This increase in GFAP expression was also measured in quantitative RT-PCR analyses (Fig 2C) from 18 hours post-laser and in Western Blot analyses only from 3 days post-laser (Fig 2E). Moreover, characteristic Iba-1 staining (Fig 2B) indicated the activation of microglial cells. While we detected only a few Iba-1-positive resting cells with long ramified processes, resident in the RGC and plexiform layers under control conditions, they greatly proliferated in response to the laser treatment, migrating into every retinal layer and differentiating into ameboid cells indicative of phagocytic microglia.

IOP elevation was also associated with strong inflammation, demonstrated by an increase in *Cd68* (Cluster of differentiation 68), *Tnfα* (Tumor necrosis factor alpha) and *Tradd* (TNFR1-associated death domain protein) gene expression in the retina of the laser-treated eyes as compared to naive or contralateral eyes (Fig 2D). This increase, initiated as early as 18h post-laser, was especially apparent for *CD68* at 3 days post-laser, and remained significant at 1 month, albeit less strong.

Immunostaining of retinal cryosections and quantitative RT-PCR were performed on both naive and contralateral eyes. No major difference was observed between these two groups regarding inflammation (Fig 2D) or gliosis (S2 Fig) indicating that disturbances induced by the laser procedure were not transferred from the treated to the contralateral eye under our experimental conditions.

The viability of RGCs was evaluated by measuring gene expression of specific markers (Brn3a, Brn3b, RBPMS and Thy-1) using quantitative RT-PCR (Fig 3A), by analysing co-staining for activated caspase3 and Brn3a on retina sections (Fig 3B), as well as by counting Brn3a + cells on flatmounted retinas (Fig 3C). Eighteen hours after laser-induced IOP elevation, we observed slight down-regulation of *Bnr3b* and *Rbpms* gene expression (-23% and -10%, respectively, *vs* naive eyes), as well as the presence of several cleaved-caspase3+/Brn3a+ cells in the RGC layer, indicating that RGCs might undergo stress leading to cell death. Consistently, we observed substantial down-regulation in gene expression of the four markers, especially *Brn3a* and *Bnr3b* (-40% and -51%, respectively, *vs* naive eyes), at 3 days post-laser. Cleaved-caspase3/ Brn3a co-staining was no longer detected in the RGC layer at this time point, suggesting that dead RGCs had been eliminated from the retina. There was an approximate 16% loss of RGCs in laser-treated compared to naive eyes, as assessed on flatmounted retinas. One month later, counting of Brn3a+ cells on flatmounted retinas indicated that about 19% of RGCs had been lost. There was no caspase3/Brn3a co-staining in the RGC layer. Unexpectedly, gene expression of the RGC markers was similar in laser-treated and non-treated eyes at this latest time point. This could be due to a compensatory overexpression of RGC specific markers by the remaining RGCs or other cell types in the retina.

## Characterization of cholesterol metabolism 18 hours post laser-induced IOP elevation

The expression of various genes implicated in cholesterol metabolism (synthesis, uptake, efflux/elimination and regulation) was measured using quantitative RT-PCR (Fig 4). As early

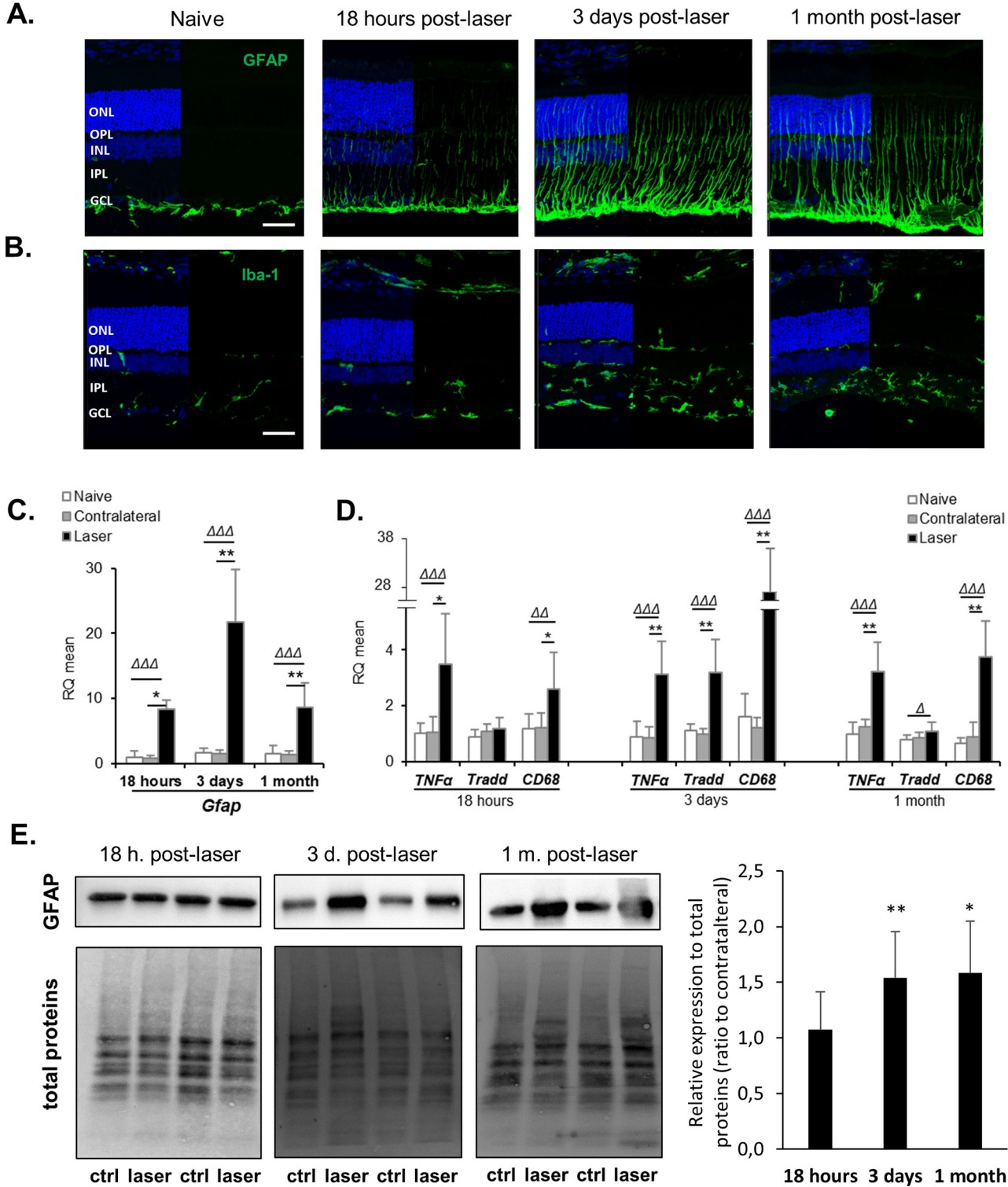

**Fig 2. Laser procedure induces persistent gliosis and strong inflammation in the retina.** Activation of macro and microglial cells was monitored by immunolabeling for GFAP (**A**) and Iba-1 (**B**), respectively. Nuclei were stained with Dapi (blue). Scale bar: 50 μm. ONL: Outer Nuclear Layer, OPL: Outer Plexiform Layer, INL: Inner Nuclear Layer, IPL: Inner Plexiform Layer, GCL: Ganglion Cell Layer. Images shown are representative of n = 6–8 retinas. (**C, D**) The expression of *Gfap* gene and genes related to inflammation was assessed using quantitative RT-PCR on retinas collected at 18 hours, 3 days or 1 month after laser photocoagulation. Data were analysed using the $2^{-\Delta\Delta CT}$ method. (**E**) The protein expression of GFAP (49 kDa) was measured by Western Blot. Results are presented as mean ± S.D, n = 6–8 retinas. Δ p<0.05, ΔΔ p<0.01, ΔΔΔ p<0.001 *vs* naive (Mann-Whitney test), * p<0.05, **p<0.01 *vs* contralateral (Wilcoxon matched-pairs signed rank test).

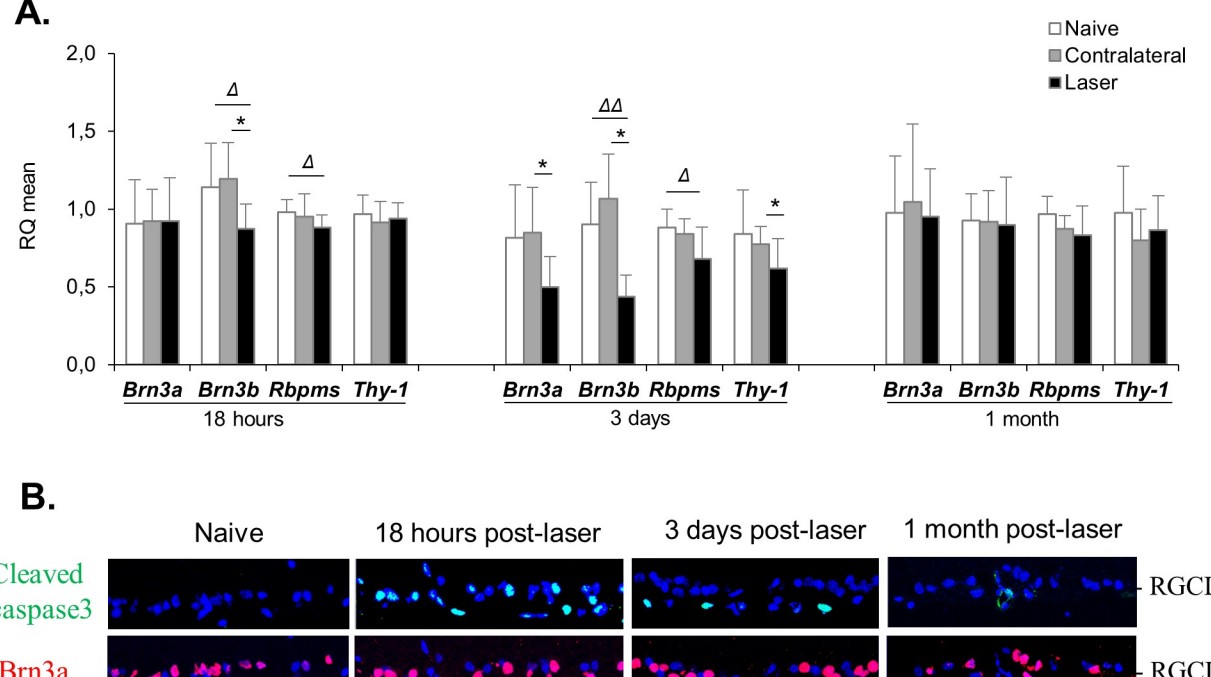

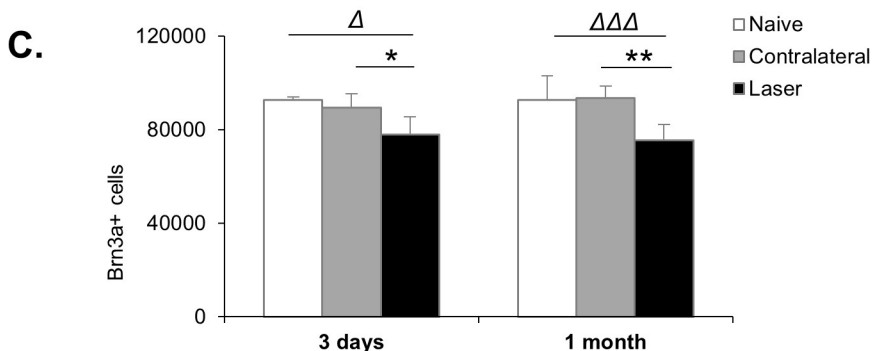

**Fig 3. Laser-induced IOP elevation triggers retinal ganglion cell death.** (A) Gene expression was assessed using quantitative RT-PCR on retinas collected at 18 hours, 3 days or 1 month after laser photocoagulation. Data were analysed using the $2^{-\Delta\Delta CT}$ method. For statistical analysis: $\Delta$ $p<0.05$, $\Delta\Delta$ $p<0.01$ *vs* naive (Mann-Whitney test), $^*$ $p<0.05$ *vs* contralateral (Wilcoxon matched-pairs signed rank test). (B) Activated cleaved-caspase 3 and Brn3a immunostaining was performed on retina cryosections showing apoptotic RGCs at 18 hours post-laser. White arrows, example of cleaved-caspase3/Brn3a co-stained cells. Images shown are representative of n = 6–8 retinas. (C) The number of RGC was assessed by counting the total number of Brn3a-positive cells on flatmounted retinas 3 days or 1 month post-laser. Results are expressed as Brn3a-positive cells / mm$^2$ and presented as mean ± S.D, n = 6 retinas for each group at 3 days, and n = 8 retinas for each group at 1 month.

as 18 hours following the laser treatment, the expression of several genes was significantly modified. As shown in Fig 4A, the transcript levels of HMGCR, the rate-limiting enzyme of cholesterol biosynthesis, was increased by 49% in the retina of laser-treated eyes, compared to naive eyes. However, the protein expression of HMGCR was unchanged on Western Blot analyses (Fig 5). The gene expression of *Ldlr* was also strongly increased (+100% compared to

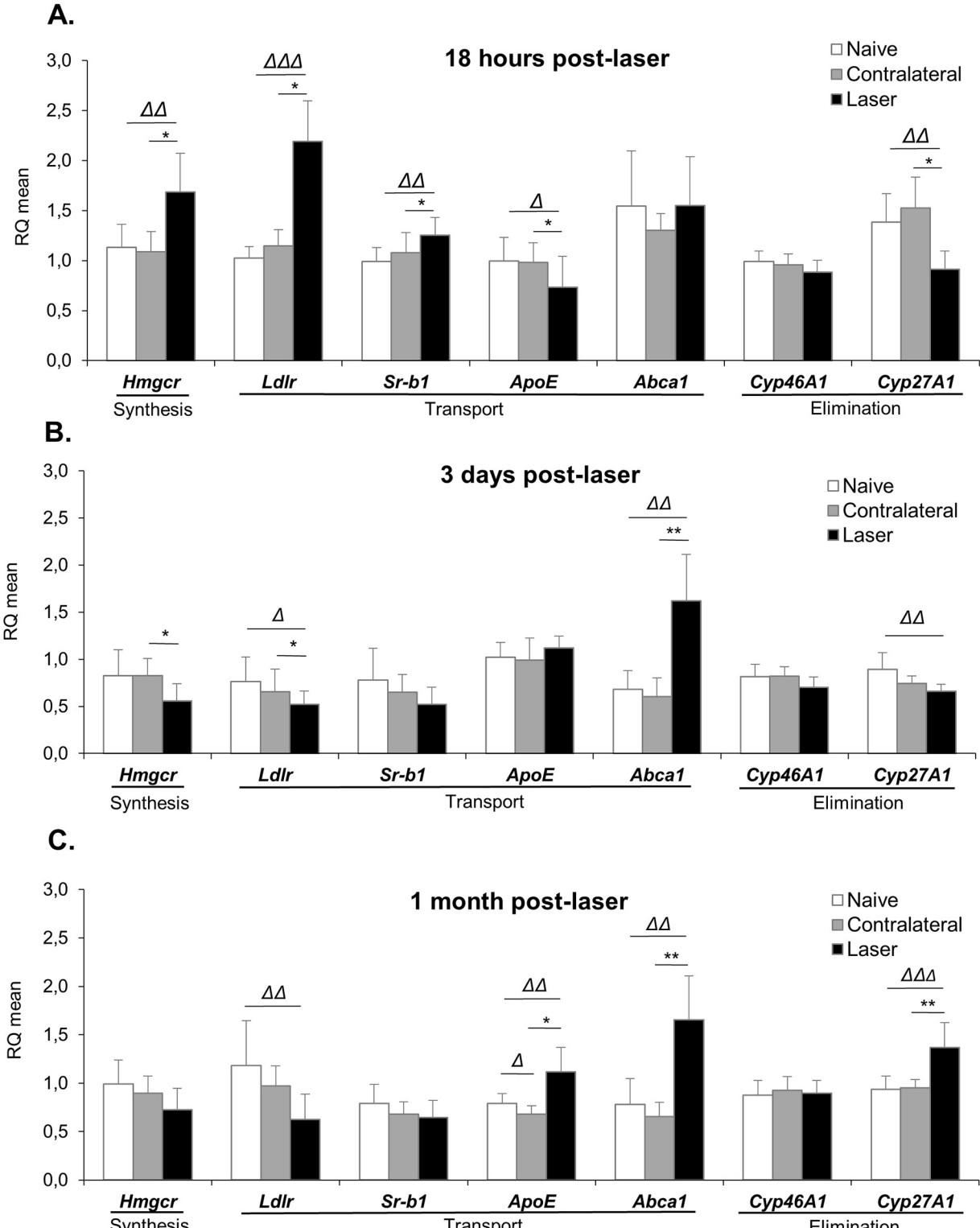

**Fig 4. Laser procedure induces dynamic modifications in the expression of genes implicated in cholesterol metabolism.** Gene expression was assessed using quantitative RT-PCR on retinas collected at 18 hours (**A**), 3 days (**B**) or 1 month (**C**) after laser photocoagulation. Data were analysed using the $2^{-\Delta\Delta CT}$ method. Results are presented as mean ± S.D, n = 6–8 retinas. Δ p<0.05, ΔΔ p<0.01, ΔΔΔ p<0.001 *vs* naive (Mann-Whitney test), * p<0.05, **p<0.01 *vs* contralateral (Wilcoxon matched-pairs signed rank test).

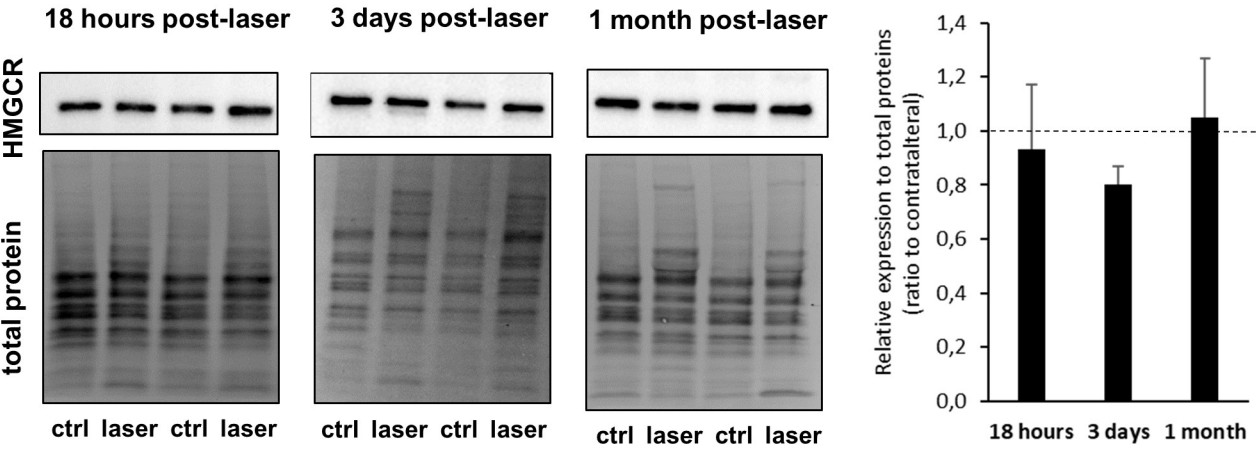

**Fig 5. Protein expression of HMGCR was unchanged in response to the laser procedure.** HMGCR (97 kDa) expression was measured using Western Blot on retinas collected at 18 hours, 3 days or 1 month after laser photocoagulation. Protein signal intensity was normalized to total protein using the stain free technology (BioRad). Representative blots are shown. Results are expressed as ratio to contralateral and represented as mean ± S. D, n = 4–7 retinas. No statistical significance with the Wilcoxon matched-pairs signed rank test.

naive eyes) and expression of *Sr-b1* was slightly increased (+26% compared to naive eyes). Conversely, transcript levels of *Apoe* were decreased by 26% in laser-treated compared to naive eyes. The gene expression of *Cyp27a1*, an enzyme playing a major role in cholesterol elimination from the retina, was also decreased by 37% in response to laser treatment. By contrast, gene expression of *Cyp46a1*, the other main cholesterol hydroxylase of the retina, appeared to be unaffected by laser treatment. Therefore, all the modifications we observed converged towards coordinated mechanisms aiming at increasing cholesterol levels.

However, at that time point, results showed that retinal cholesterol levels were not different between laser-treated and non-treated eyes (Fig 6A). Two major precursors of cholesterol, desmosterol and lathosterol, were also quantified and did not show any change (Fig 6B). Regarding oxysterols, the levels of 24S-OHC were similar between laser-treated and non-treated eyes (Fig 6C). Neither 27-OHC nor 27-COOH, its metabolite, could be detected.

## Characterization of cholesterol metabolism 3 days post laser-induced IOP elevation

Contrary to the observations made at the earlier 18 hours post-laser time-point, the levels of cholesterol precursors, desmosterol and particularly lathosterol, were strongly increased in the retina of laser-treated eyes as compared to controls three days after laser treatment (Fig 6B) (+40% and +194% compared to naive, respectively). The levels of cholesterol were slightly but significantly increased (+14% compared to naive eyes) (Fig 6A). The levels of the oxysterol 24S-OHC were unaffected by the laser procedure at that time point (Fig 6C). Again, neither 27-OHC nor 27-COOH could be detected.

Regarding the expression of genes implicated in cholesterol metabolism (Fig 4B), results showed a reversion of the modifications observed at 18 hours post-laser. Indeed, the expression of *Hmgcr* gene was significantly decreased in the retina of laser-treated as compared to non-treated eyes while we were not able to detect any significant modification in HMGCR expression by Western Blot (Fig 5). *Ldlr* gene expression was also decreased (-32% for both compared to naive eyes), as well as the expression of *Sr-b1* gene without reaching statistical

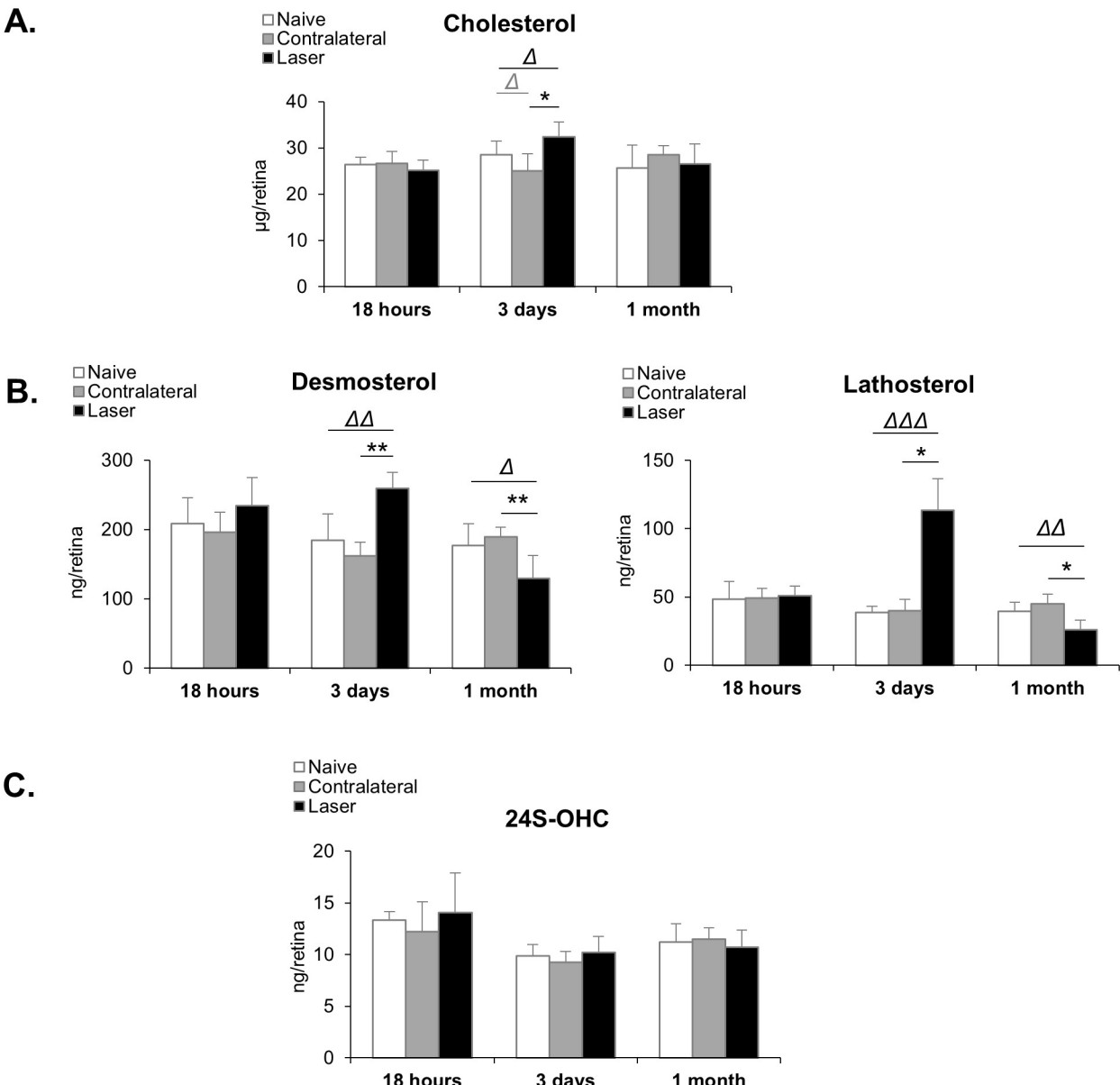

**Fig 6. Laser procedure induces a transient increase in cholesterol and cholesterol precursor levels in the retina. (A)** Cholesterol was quantified using GC-FID. Cholesterol precursor levels, desmosterol and lathosterol **(B)**, and levels of 24S-hydroxycholesterol (24S-OHC) **(C)** were measured using GC-MS. Results are presented as mean ± S.D. Δ $p < 0.05$, ΔΔ $p < 0.01$, ΔΔΔ $p < 0.001$ *vs* naive (Mann-Whitney test), * $p < 0.05$, **$p < 0.01$ *vs* contralateral (Wilcoxon matched-pairs signed rank test). n = 7–8 retinas in each group.

significance. The gene expression of *Apoe* was similar in laser- and non-treated eyes and the expression of its partner, *Abca1* was strongly increased (+137% compared to naive eyes). These changes are in line with activation of mechanisms involved in decreasing retinal cholesterol levels, except for *Cyp27a1*, whose transcript levels were still decreased in the laser-treated compared to naive eyes (-26% *vs* naive). Again, the expression of *Cyp46a1* gene was unaffected by the intervention.

## Characterization of cholesterol metabolism 1 month post laser-induced IOP elevation

At the later time point of 1 month post-laser, retinal cholesterol levels of treated eyes were back to the levels of non-treated eyes (Fig 6A). The levels of the cholesterol precursors, desmosterol and lathosterol, were decreased in laser-treated as compared to non-treated eyes (-27% and -34% vs naive, respectively) (Fig 6B). The levels of 24S-OHC were unchanged (Fig 6C).

The expression of genes implicated in cholesterol metabolism between treated and non-treated eyes was mostly similar at 3 days and 1 month post-laser (Fig 4C). The expression of *Hmgcr* gene was decreased (-27% compared to naive eyes), although not significantly. Again, we could not detect any change in the protein levels of HMGCR using Western Blot (Fig 5). The expression of *Ldlr* gene was decreased (-47%), while the expression of *Abca1* gene was increased (+ 112%). In addition, the transcript levels of *Apoe* and *Cyp27a1* were significantly increased in the retina of laser-treated as compared to non-treated eyes (+41% and +46% compared to naive, respectively). Similarly to what happened at the previous time points, the expression of *Cyp46a1* gene appeared to be unaffected by the laser treatment. Overall, these results indicate that the hypocholesterogenic mechanisms activated by 3 days post-laser were maintained and even reinforced 1 month later.

## Discussion

Our experimental model of ocular hypertension is characterized by inflammation, gliosis and RGC death, as previously described [16, 18, 21]. The major finding of the present study is that this model is also associated with early transient alterations in cholesterol metabolism (Fig 7).

While the mechanisms responsible for such alterations remain to be determined, several hypotheses can be considered. First, a link between increased IOP and changes in the metabolism of cholesterol has been reported in a couple of studies focusing on the CYP46A1 enzyme. The first one was performed in *ex vivo* rat retina, and the authors showed that IOP elevation increased the expression of *Cyp46a1* gene and protein as well as the levels of 24S-OHC, while decreasing cholesterol levels [15]. The second study was performed in our laboratory in a rat model of laser-induced IOP elevation [16]. A transient increase in the retinal expression of CYP46A1 enzyme was measured 3 days following the laser procedure, as well as an increase in 24S-OHC levels peaking at 1 month. This was associated with retinal gliosis and systemic inflammation as exemplified by increased MCP-1 and ICAM-1 plasma levels. On the contrary, in the present work, *Cyp46a1* gene expression and the 24S-OHC levels were unchanged at each experimental time point tested. This result is consistent with a study published by Ohyama et al. indicating that the *Cyp46a1* gene expression is very stable and only weakly regulated at the transcriptional level [22]. The constitutive expression of the enzyme in neurons suggests that CYP46A1 activity is essential for their survival. This concept is strengthened by the fact that CYP46A1 inhibition has been reported to lead to neuronal death [23]. Second, regarding inflammation and gliosis, several studies suggest that hypercholesterolemia and cholesterol derivatives could generate neuro-inflammation, notably associated with microglial activation [24–27]. There are only sparse data to support the idea that, conversely, inflammation and gliosis could result in disturbances of cholesterol metabolism. In a mouse model of Alzheimer's disease, Orre et al. described strong alterations in the expression of many cholesterol-related genes, specifically in activated macroglial cells (astrocytes), and microglia [28]. These observations are consistent with other studies suggesting the leading role of glial cells in brain cholesterol metabolism [29–31]. In the retina, there is also evidence that Müller cells are major players of cholesterol metabolism [2–4], especially of cholesterol biosynthesis and efflux

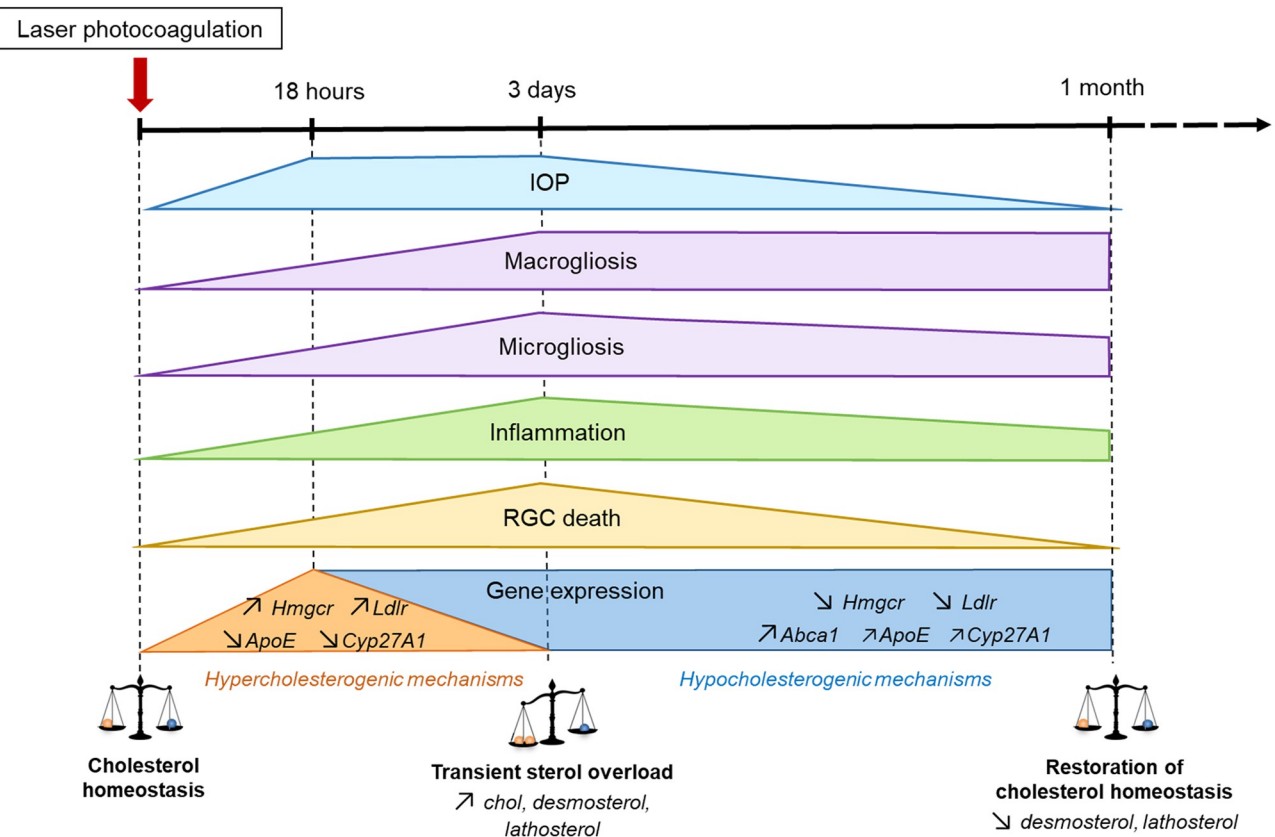

**Fig 7. A dynamic scheme summarizing the changes observed in our ocular hypertensive model.** The laser procedure induced increased IOP, inflammation, activation of glial cells and RGC stress and death. Concomitantly, a transient increase in the retinal content of cholesterol and precursors occurred, followed by a restoration of cholesterol homeostasis associated with modifications of the gene expression of major players of cholesterol metabolism.

via ApoE and ABCA1. Using primary cell cultures, we have shown that Müller cells express the necessary machinery for cholesterol synthesis, uptake and export via lipoproteins and are able to adjust these pathways [4]. Since microglia also express the machinery related to cholesterol metabolism, especially for efflux, they are most likely active players as well [32]. It is therefore reasonable to think that the activation of both macro and microglia, observed in our model, could actively participate in the modifications of cholesterol metabolism we measured. Third, similarly to what has been proposed in the brain, cholesterol homeostasis in the retina might rely on close cooperation between glial cells acting as cholesterol providers and, neurons as cholesterol consumers [33]. One can thus expect that the RGC stress we observed from 18 hours to 3 days post-laser could also generate some changes in cholesterol metabolism. However, our *in vivo* model does not permit to decipher precisely which cell(s) contribute to which modifications of the cholesterol metabolism. Indeed, the modifications we observe for the expression of cholesterol-related genes or the sterol levels reflect the retina as a whole.

Interestingly, it appeared that compensatory mechanisms were activated 3 days post-laser in response to the early disturbances in cholesterol metabolism leading to the restoration of cholesterol homeostasis by one month. Literature data tend to indicate that cholesterol overload might be neurotoxic [23, 34] and that the recovery of cholesterol status displays a

neuroprotective effect, as has been shown by CYP46A1 overexpression in a Huntington's disease mouse model [35]. We therefore hypothesize that hypocholesterogenic mechanisms may prevent side effects of abnormal sterol levels and limit the extent of RGC death, which is consistent with the moderate loss of RGCs we observed in our experimental model of glaucoma (about 20% of death one month after IOP elevation). Conversely, it is also likely that the moderate RGC stress occurring here enabled cholesterol homeostasis to recover, which might not be the case in a more drastic experimental model, as performed by Salinas-Navarro et al. and Shnebelen et al. (60% RGC loss 3 weeks post-laser and 68% after 3 months, respectively) [18, 36]. The variability in the severity of RGC or axonal loss and IOP profile between laboratories might be due to laser specifications and technical use: spot size and number, power, use of slit lamp or endoscopic probes. . . Similar experiments will have to be performed in other glaucoma models in order to confirm the role of cholesterol homeostasis in RGC viability.

We show here a solid transcriptional regulation of the major players of cholesterol homeostasis in the retina and it is striking that a panel of genes implicated in firstly hyper, and subsequently hypo-cholesterogenic mechanisms were modulated in a coordinated manner (Fig 7). Our results lead to think that, in our model of laser-induced IOP elevation, a common regulator of cholesterol metabolism could be modulated. SREBP2 could be this regulator since it is known to upregulate the expression of cholesterogenic genes, especially as its main targets are *Hmgcr* and *Ldlr* coding genes [37], which are strongly regulated in our model. This contrasts with the observations made by Zheng et al. who reported a lack of transcriptional responsiveness in their experimental models and suggested that the SREBP2 pathway of transcriptional regulation is not operative in the retina. Indeed, using high cholesterol diet-fed or simvastatin-treated mice, the authors reported that changes in sterol levels (cholesterol and precursors) did not significantly affect the transcription of retinal cholesterol-related genes [38]. LXRα is a transcription factor known to activate hypo-cholesterogenic mechanisms. It might be implicated as well in our experimental model since *Apoe* and *Abca1*, highly regulated under our conditions, are among its major targets [39]. The robust regulation of *Ldlr*, *Apoe* and *Abca1* gene expression we observed in our model indicates that lipoprotein trafficking is very responsive to transcriptional regulation and that it might be a major way to maintain cholesterol homeostasis in the retina, as we already proposed on the basis of previous observations made on primary rat Müller cell cultures [4]. While neither gene expression of *Cyp46a1* nor levels of 24S-OHC were altered in our experimental model, *Cyp27a1* gene expression was initially downregulated following laser treatment, and then upregulated. However, we could not detect 27-COOH in the rat retina and cannot therefore conclude on the regulatory importance of this pathway in our model. Considering our limit of detection for 27-COOH, we can estimate that levels are below 0.75 pmol / mg prot in our rat retina samples. This is in accordance with the observations of Omarova et al. on mouse retinas [40] since they could not detect 27-COOH with a limit of detection of 0.5 pmol / mg protein, but not with a previous study of Saadane et al. reporting detectable levels of 27-COOH in mouse retina (2 pmol / mg protein) [32]. Another study also reported significant levels of 27-COOH in post-mortem bovine and human retinas [41]. This discrepancy could be due to species specific differences or to the time elapsed between death of the animal and retinal collection since the latter study indeed showed an increase in retinal 27-COOH content during this period.

Our results do not permit to claim that the modifications of sterol levels we measured result from the gene expression changes observed. First, and surprisingly, the modifications of *Hmgcr* gene expression did not translate into detectable modifications of protein expression. Protein expression was measured by Western Blot and one cannot rule out that this semi-quantitative technique did not enable to measure slight changes in protein levels (as it was the case for GFAP at 18 hours post-laser: an increased expression could be measured using

quantitative RT-PCR and immunofluorescence but not western blot). However, it is also possible that the changes we measured in gene expression using RT-qPCR (less than two-fold) were not sufficient to significantly modify protein levels or were compensated by other regulatory mechanisms. While transcriptional regulation of cholesterol metabolism by SREBP is the most well-described mechanism, regulation of HMGCR degradation has also been shown, via the action of the protein Insig. It has also been shown that HMGCR translation can be regulated even though it has been much less investigated [42, 43]. Second, sterol levels do not seem to directly correlate with gene expression of the major players of cholesterol biosynthesis, transport and efflux at each time point tested. This could be explained by a lag phase for the changes in gene expression to affect the levels of cholesterol and precursors. However, restoration of cholesterol homeostasis and decreased levels of precursors at 1 month post-laser, especially desmosterol which is a LXR activator, are not consistent with the fact that *Apoe*, *Abca1* and *Cyp27A1* gene levels are still increased.

It remains that the modifications in lathosterol and desmosterol levels we measured (increase at 3 days and decrease at 1 month post-laser), leading to a slight and transient cholesterol overload, are indicative of a modulation of cholesterol biosynthesis and likely of HMGCR activity. If not linked to gene or protein expression changes, this could result from post-translational modifications. Indeed, strong post-translational regulation mechanisms have been described for HMGCR, whose activity can be regulated by phosphorylation [42]. While HMGCR is usually targeted, as the rate-limiting enzyme of the cholesterol biosynthesis pathway, other enzymes that we did not investigate could also be implicated. Most of the enzymes of cholesterol biosynthetic pathway have actually been shown to be regulated by SREBP at the gene level [44] while the squalene synthase has been shown to be responsive to LXRα [45]. Epigenetic regulation such as histone acetylation has also been described for lanosterol synthase gene expression [46].

## Conclusion

On one hand our study reveals that ocular hypertension may be associated with transient alterations in cholesterol metabolism. On the other hand, it reveals that the retina is able to restore cholesterol homeostasis under stress conditions. A transcriptional regulation of the major players of cholesterol metabolism clearly occurs in the retina but the importance of this mechanism in the maintenance of cholesterol homeostasis in this tissue remains unclear.

## Supporting information

**S1 Fig. Laser procedure induces a prolonged ocular hypertension.** The right eye of Sprague-Dawley rats was subjected to laser photocoagulation of the trabecular meshwork, episcleral veins and limbal plexus. The left eye was considered as contralateral eye. The intraocular pressure (IOP) was monitored regularly in both eyes as well as on naive eyes (rats which were not subjected to the laser procedure), under gas anaesthesia, with a rebound tonometer (Icare®TonoLab). Results are presented as mean ± SEM of a minimum of 25 animals. Δ $p<0.05$, ΔΔ $p<0.01$, ΔΔΔ $p<0.001$ *vs* naive (unpaired t test). * $p<0.05$, ** $p<0.01$, *** $p<0.001$ *vs* contralateral (paired t test).
(DOCX)

**S2 Fig. Glial activation was not detected in the contralateral retinas in response to laser procedure.** Immunostaining for GFAP and Iba-1 was performed on retinal cryosections of naive and contralateral eyes. No major difference was observed between these two untreated groups at any time point regarding activation of macro and microglial cells under our

experimental conditions. Scale bar: 50 μm. Images shown are representative of n = 6–8 retinas at 18 hours, 3 days and 1month post-laser.
(DOCX)

**S1 Table. References of TaqMan assays (Applied Bioscience) used for QRTPCR analyses.**
ApoE: Apolipoprotein E, Cd68: Cluster of Differentiation 68, Cyp27a1: cytochrome P450 family 27 subfamily A member 1, Cyp46a1: cytochrome P450 family 46 subfamily A member 1, Gfap: Glial fibrillary acidic protein, Hmgcr: 3-hydroxy-3-methylglutaryl-CoA reductase, Nr1h2: nuclear receptor subfamily 1 group H member 2, Nr1h3: nuclear receptor subfamily 1 group H member 3, Pou4f1: POU Domain, Class 4, Transcription Factor 1, Pou4f2: POU Domain, Class 4, Transcription Factor 2, Rbpms: RNA-binding protein with multiple splicing, Scarb1: scavenger receptor class B member 1, Srebf2: Sterol regulatory element-binding transcription factor 2, Thy1: Thymocyte differentiation antigen 1, Tnf: Tumor necrosis factor, Tradd: Tumor necrosis factor receptor type 1-associated death domain protein.
(DOCX)

**S1 Raw images.**
(PDF)

# Acknowledgments

The authors are grateful to the animal facility of Centre des Sciences du Goût et de l'Alimentation (CSGA) for animal care. We thank Dimacell platform and Christine Arnould for technical assistance regarding confocal microscopy. We also thank Claire Chabanet of CSGA for technical assistance regarding statistical analysis of the data. English spelling and grammar were verified by a proofreading company.

# Author Contributions

**Conceptualization:** Elise Léger-Charnay, Elodie A. Y. Masson.

**Formal analysis:** Elise Léger-Charnay, Elodie A. Y. Masson.

**Funding acquisition:** Ségolène Gambert, Lionel Bretillon.

**Investigation:** Elise Léger-Charnay, Ségolène Gambert, Lucy Martine, Elisabeth Dubus, Marie-Annick Maire, Bénédicte Buteau, Tristan Morala, Elodie A. Y. Masson.

**Methodology:** Lucy Martine, Elisabeth Dubus, Alain M. Bron, Elodie A. Y. Masson.

**Project administration:** Ségolène Gambert, Lionel Bretillon, Elodie A. Y. Masson.

**Software:** Vincent Gigot.

**Supervision:** Alain M. Bron, Lionel Bretillon, Elodie A. Y. Masson.

**Validation:** Elise Léger-Charnay, Lucy Martine, Elisabeth Dubus, Bénédicte Buteau, Lionel Bretillon, Elodie A. Y. Masson.

**Visualization:** Elise Léger-Charnay, Elisabeth Dubus.

**Writing – original draft:** Elise Léger-Charnay, Elodie A. Y. Masson.

**Writing – review & editing:** Ségolène Gambert, Lucy Martine, Elisabeth Dubus, Marie-Annick Maire, Bénédicte Buteau, Tristan Morala, Vincent Gigot, Alain M. Bron, Lionel Bretillon, Elodie A. Y. Masson.

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
