## [Decision Letter · Decision Letter 0]

4 Nov 2021

PONE-D-21-27664Retinal cholesterol metabolism is perturbated in response to experimental glaucoma in the ratPLOS ONE

Dear Dr. MASSON,

Thank you for submitting your manuscript to PLOS ONE. After careful consideration, we feel that it has merit but does not fully meet PLOS ONE’s publication criteria as it currently stands. Therefore, we invite you to submit a revised version of the manuscript that addresses the points raised during the review process.

The reviews for you manuscript were received and, as you can see, they appear quite divergent. The major concern of reviewer 2 is the lack of clarity in explaining the proposed physiopathology.

I believe most concerns can be addressed by better explaining the link between markers for inflammation and laser-induced intra-ocular hypertension. However additional controls would be required to document changes in genes related to cholesterol metabolism (as identified by RT-PCR), as suggested by Reviewer 2.

We look forward to receiving your revised manuscript.

Kind regards,

Tudor C Badea, M.D., M.A., Ph.D.

Academic Editor

PLOS ONE

Journal Requirements:

2. Please include the full name of the IRB or ethics committee who approved or waived your study.

In your cover letter, please note whether your blot/gel image data are in Supporting Information or posted at a public data repository, provide the repository URL if relevant, and provide specific details as to which raw blot/gel images, if any, are not available. Email us at plosone@plos.org if you have any questions

4. We note that Figure 1 in your submission contain copyrighted images. All PLOS content is published under the Creative Commons Attribution License (CC BY 4.0), which means that the manuscript, images, and Supporting Information files will be freely available online, and any third party is permitted to access, download, copy, distribute, and use these materials in any way, even commercially, with proper attribution. For more information, see our copyright guidelines: http://journals.plos.org/plosone/s/licenses-and-copyright.

    1. You may seek permission from the original copyright holder of Figure(s) [#] to publish the content specifically under the CC BY 4.0 license.

Reviewers' comments:

Reviewer's Responses to Questions

**Comments to the Author**

1. Is the manuscript technically sound, and do the data support the conclusions?

Reviewer #1: Yes

Reviewer #2: No

2. Has the statistical analysis been performed appropriately and rigorously? 

Reviewer #1: Yes

Reviewer #2: Yes

3. Have the authors made all data underlying the findings in their manuscript fully available?

Reviewer #1: Yes

Reviewer #2: Yes

4. Is the manuscript presented in an intelligible fashion and written in standard English?

Reviewer #1: Yes

Reviewer #2: Yes

5. Review Comments to the Author

Reviewer #1: The authors have characterized cholesterol metabolism perturbations in a rat laser-induced glaucoma model. The manuscript is well written, and the study provides valuable information about retinal metabolic changes resulting from elevated pressure (in the rodent model at least).

The methods are well described, and the techniques are all well-established in the literature.

The dynamic changes in cholesterol metabolism have been well demonstrated. The reduced expression of Cyp27a1, in response to elevated IOP treatment, but not Cyp46a1, is a most interesting finding.

The authors provide a thoughtful discussion of the possible underlying mechanisms.

Conclusions are presented in an appropriate fashion and are supported by the data.

The Figures are of good quality and the schematics helpful.

Specific queries

There is one very curious result that needs some explanation

The authors have measured RGC injury using a combination of RGC-specific mRNA quantification with qPCR and with RGC counts using Brn3a-immunostained retinal flat mounts.

Fig 3A shows early loss of Brn3a message with recovery by 1 month. The RBPMS and Thy-1 RQ means also recover.

Curiously, the flat mount data (Fig 3C) show a relatively modest but increasing loss of Brn3a immunoreactive cells out to 1 month.

How is the mRNA data at 1 month reconciled with the flat mount data?

Reviewer #2: The paper „Retinal cholesterol metabolism is perturbated in response to experimental glaucoma in the rat“ is an intersting topic. In conclusion, they report that ocular hypertension is associated with transient major dynamic changes in retinal cholesterol metabolism.

However, it is not clearly written and confusing as many results are shown such as Iba-1 , GFAP , Caspase changes, that don`t have much to do with the cholesterol changes. It is not even explained how they could be associated with those changes. The findings they had regarding cholesterol metabolism were only „As early as 18 hours after the laser procedure, genes implicated in cholesterol biosynthesis and uptake were upregulated (+49 % and +100 % for Hmg-CoA reductase and Ldlr genes respectively, vs. naive eyes) while genes involved in efflux were downregulated (-26 % and -37 % for Apoe and Cyp27a1 genes, respectively). „ No furhter validation of those changes was done , e.g. IHC, Proteoics, WB or no furhter explanation oft he pathomechanims. All other findings are descriptive and the authors fail to explain the association.

It is a shame as the storyline is not obvious and loads of findings are reported which might be of interesting value but the association is not clear.

6. PLOS authors have the option to publish the peer review history of their article (what does this mean?). If published, this will include your full peer review and any attached files.

Reviewer #1: **Yes: **Robert Casson

Reviewer #2: No

---

## [Author Response · Author response to Decision Letter 0]

13 Dec 2021

PONE-D-21-27664

Retinal cholesterol metabolism is perturbated in response to experimental glaucoma in the rat

PLOS ONE

Dear Dr. MASSON,

Thank you for submitting your manuscript to PLOS ONE. After careful consideration, we feel that it has merit but does not fully meet PLOS ONE’s publication criteria as it currently stands. Therefore, we invite you to submit a revised version of the manuscript that addresses the points raised during the review process.

The reviews for you manuscript were received and, as you can see, they appear quite divergent. The major concern of reviewer 2 is the lack of clarity in explaining the proposed physiopathology.

I believe most concerns can be addressed by better explaining the link between markers for inflammation and laser-induced intra-ocular hypertension. However additional controls would be required to document changes in genes related to cholesterol metabolism (as identified by RT-PCR), as suggested by Reviewer 2.

Dear Dr Badea,

We indeed noticed the divergence between the two reviewers’ point of view. The main criticism of Reviewer #2 is that there is “no further explanation of the pathomechanisms”. We acknowledge this is a rather descriptive study. Obviously, and as usual, the results raise questions, so far unresolved. Specifically, the physiopathologic mechanisms explaining the link between increased IOP, gliosis, inflammation, changes in cholesterol metabolism and RGC death are still unknown. Several hypotheses regarding the potential role of cholesterol metabolism in our pathological model had already been developed in the discussion of the original version of the manuscript, as underlined by Reviewer #1. To gain clarity in this proposed physiopathology, we designed a scheme that we submit to Reviewer #2. Regarding the absence of a protein validation of the changes in gene expression we measured, we acknowledge it as a limitation of our work. We had tested antibodies using Western Blot for ABCA1 and LDLR but we decided not to utilize the poor-quality results obtained to avoid depreciating the quality of the manuscript. We hope you will consider our study provide sufficient amount of reliable data for a publication in PLOS ONE and that it is worth making our findings accessible to the scientific community despite their descriptive nature. 

A pdf file (named S1_raw_images) that contains all the original Western Blot images for GFAP and HMGCR, uncropped and unadjusted, underlying the manuscript results was created according to PLOSONE guidelines. It was uploaded as a Supporting Information file. Preparing this file, we realized that, in Figure 5, the images for HMGCR 3d and 1 month post-laser had not been flipped for a presentation in un intuitive order (ctrl before laser) and did not therefore correspond to the legend. This mistake was corrected in a new version of Figure 5. It did not affect the quantification of the bands that had been properly done on raw data.

We obtained permission to publish Figure 1 from the copyright holder (Biorender). The document was uploaded with the submission of the revised manuscript.

Reviewers' comments:

Reviewer's Responses to Questions

Comments to the Author

1. Is the manuscript technically sound, and do the data support the conclusions?

Reviewer #1: Yes

Reviewer #2: No

2. Has the statistical analysis been performed appropriately and rigorously? 

Reviewer #1: Yes

Reviewer #2: Yes

3. Have the authors made all data underlying the findings in their manuscript fully available?

Reviewer #1: Yes

Reviewer #2: Yes

4. Is the manuscript presented in an intelligible fashion and written in standard English?

Reviewer #1: Yes

Reviewer #2: Yes

5. Review Comments to the Author

Reviewer #1: The authors have characterized cholesterol metabolism perturbations in a rat laser-induced glaucoma model. The manuscript is well written, and the study provides valuable information about retinal metabolic changes resulting from elevated pressure (in the rodent model at least).

The methods are well described, and the techniques are all well-established in the literature.

The dynamic changes in cholesterol metabolism have been well demonstrated. The reduced expression of Cyp27a1, in response to elevated IOP treatment, but not Cyp46a1, is a most interesting finding.

The authors provide a thoughtful discussion of the possible underlying mechanisms.

Conclusions are presented in an appropriate fashion and are supported by the data.

The Figures are of good quality and the schematics helpful.

Specific queries

There is one very curious result that needs some explanation

The authors have measured RGC injury using a combination of RGC-specific mRNA quantification with qPCR and with RGC counts using Brn3a-immunostained retinal flat mounts.

Fig 3A shows early loss of Brn3a message with recovery by 1 month. The RBPMS and Thy-1 RQ means also recover.

Curiously, the flat mount data (Fig 3C) show a relatively modest but increasing loss of Brn3a immunoreactive cells out to 1 month.

How is the mRNA data at 1 month reconciled with the flat mount data?

We thank reviewer #1 for his interest in our study and his appreciation of the manuscript. We agree that the recovery of the gene expression of RGC specific markers is most intriguing. The hypothetical explanation we can propose is that the expected decrease of qPCR signal due to the loss of RGC, as measured on flatmounted retinas at 1 month post-laser, was compensated by an overexpression of these same markers by the remaining RGC or other cell types in the retina. We added this point to the corresponding Result section (p 16).

Reviewer #2: The paper „Retinal cholesterol metabolism is perturbated in response to experimental glaucoma in the rat“ is an intersting topic. In conclusion, they report that ocular hypertension is associated with transient major dynamic changes in retinal cholesterol metabolism.

However, it is not clearly written and confusing as many results are shown such as Iba-1 , GFAP , Caspase changes, that don`t have much to do with the cholesterol changes. It is not even explained how they could be associated with those changes. The findings they had regarding cholesterol metabolism were only „As early as 18 hours after the laser procedure, genes implicated in cholesterol biosynthesis and uptake were upregulated (+49 % and +100 % for Hmg-CoA reductase and Ldlr genes respectively, vs. naive eyes) while genes involved in efflux were downregulated (-26 % and -37 % for Apoe and Cyp27a1 genes, respectively). „ No furhter validation of those changes was done , e.g. IHC, Proteoics, WB or no furhter explanation oft he pathomechanims. All other findings are descriptive and the authors fail to explain the association.

It is a shame as the storyline is not obvious and loads of findings are reported which might be of interesting value but the association is not clear.

We thank reviewer #2 for his interesting comments on our manuscript and acknowledge the lack of clarity and explanation of the physiopathologic mechanisms.

Reviewer #2 raises the fact that many results don’t have much to do with the cholesterol changes. Indeed, the results shown in Figures 2 (gliosis and inflammation) and 3 (RGC death) aim to characterize our experimental model of glaucoma. We presented these results in order to reveal the potential triggers and/or consequences of the changes in cholesterol metabolism we report in this specific model. We changed the title of the corresponding Result section to “characterization of the experimental model of glaucoma” to clarify the utility of these data (p 14).

The lack of explanation of the pathomechanisms is also a matter of criticism. At that stage of the study, we cannot explain the association between the increase in IOP, inflammation and gliosis we observed and the changes in cholesterol metabolism we measured. It is an exploratory work since no similar study had been performed so far. We made several hypotheses in the discussion section based on literature data to explain the potential link between cholesterol metabolism and i) increased IOP (lines 397-410), ii) inflammation and gliosis (lines 410-423), iii) RGC death (424-430). To make this clearer, we designed a scheme with a legend (see below). In this scheme, we also illustrate the idea that the restoration of cholesterol homeostasis might be protective against RGC death. If reviewer #2 considers it might be useful for the readers and not too speculative, we could include it to the final version of the manuscript, possibly in replacement of Figure 7. The abstract was also modified to make the potential association between the different observations clearer.

Fig.8. A proposed model for the potential role of retinal cholesterol metabolism in our ocular hypertensive model of glaucoma. The laser procedure induced increased IOP, Inflammation and gliosis. Consecutively, the activation of hypercholesterogenic mechanisms resulted in transient retinal hypercholesterolemia known to be toxic for neurons. Activation of hypocholesterogenic feed back mechanisms then enabled the restoration of cholesterol homeostasis and prevented more pronounced RGC death. Arrows represent the potential causal links between the events based on our observations and on commonly accepted pathways. Dashed arrows correspond to undetermined mechanisms.

Regarding the absence of validation of the measured changes in gene expression by measuring protein expression, we agree with reviewer #2 it is a limitation of our work. We did it for HMGCR using Western Blot (Fig. 5) and did not measure any change in protein expression despite the significant changes in gene expression as measured by RT-qPCR, maybe due to post-translational regulations. We also investigated the 2 other players whose gene expression was strongly regulated: ABCA1 and LDLR. Unfortunately, the antibodies tested under various conditions generated blots of poor quality that we considered insufficient to provide reliable results. As one can see below, the signal of bands at the expected molecular weight was faint while we detected many aspecific bands. However, the coordinated regulation of gene expression we observed at the different time points combined with a coherent modification in sterol levels gives us confidence in the fact that a regulation of cholesterol metabolism is actually involved in our experimental model of glaucoma.

---

## [Decision Letter · Decision Letter 1]

17 Feb 2022

Retinal cholesterol metabolism is perturbated in response to experimental glaucoma in the rat

PONE-D-21-27664R1

Dear Dr. MASSON,

We’re pleased to inform you that your manuscript has been judged scientifically suitable for publication and will be formally accepted for publication once it meets all outstanding technical requirements.

Kind regards,

Tudor C. Badea, M.D., M.A., Ph.D.

Academic Editor

PLOS ONE

Additional Editor Comments (optional):

Reviewers' comments:

Reviewer's Responses to Questions

**Comments to the Author**

1. If the authors have adequately addressed your comments raised in a previous round of review and you feel that this manuscript is now acceptable for publication, you may indicate that here to bypass the “Comments to the Author” section, enter your conflict of interest statement in the “Confidential to Editor” section, and submit your "Accept" recommendation.

Reviewer #1: All comments have been addressed

2. Is the manuscript technically sound, and do the data support the conclusions?

Reviewer #1: Yes

3. Has the statistical analysis been performed appropriately and rigorously? 

Reviewer #1: Yes

4. Have the authors made all data underlying the findings in their manuscript fully available?

Reviewer #1: Yes

5. Is the manuscript presented in an intelligible fashion and written in standard English?

Reviewer #1: Yes

6. Review Comments to the Author

Reviewer #1: (No Response)

7. PLOS authors have the option to publish the peer review history of their article (what does this mean?). If published, this will include your full peer review and any attached files.

Reviewer #1: **Yes: **Robert James Casson

---

## [Editor Report · Acceptance letter]

3 Mar 2022

PONE-D-21-27664R1 

Retinal cholesterol metabolism is perturbated in response to experimental glaucoma in the rat 

Dear Dr. MASSON:

I'm pleased to inform you that your manuscript has been deemed suitable for publication in PLOS ONE. Congratulations! Your manuscript is now with our production department. 

Kind regards, 

on behalf of

Dr. Tudor C. Badea 

Academic Editor

PLOS ONE